# Medical Image Segmentation via Unsupervised Convolutional Neural Network

**Junyu Chen**          JCHEN245@JHMI.EDU  and  **Eric C. Frey**          EFREY@JHMI.EDU

*Department of Radiology and Radiological Science, Johns Hopkins Medical Institutes, USA*

*Department of Electrical and Computer Engineering, Johns Hopkins University, USA*

## Abstract

For the majority of the learning-based segmentation methods, a large quantity of high-quality training data is required. In this paper, we present a novel learning-based segmentation model that could be trained semi- or un- supervised. Specifically, in the unsupervised setting, we parameterize the Active contour without edges (ACWE) framework via a convolutional neural network (ConvNet), and optimize the parameters of the ConvNet using a self-supervised method. In another setting (semi-supervised), the auxiliary segmentation ground truth is used during training. We show that the method provides fast and high-quality bone segmentation in the context of single-photon emission computed tomography (SPECT) image. The source code of this work is available online at https://github.com/junyuchen245/Unsuprevised_Seg_via_CNN/

**Keywords:** image segmentation, active contour, convolutional neural network.

## 1. Introduction

A great deal of recent work in image segmentation has been based on deep neural networks, which require a large amount of accurately annotated training data. However, generating annotated data is a time-consuming process, which could take several months to a year to complete (Segars et al., 2013). By contrast, conventional methods, such as clustering (Chen et al., 2019a) and level-set (Chan and Vese, 2001) based techniques, solely rely on the statistics of intensities in a given image. Although they do not require any training data, the intensity statistics have to be remodeled for every input image, resulting in an significant computation burden. In this work, we take the advantage of the best of both methods and propose a learning-based method that can be trained in a semi-supervised or unsupervised manner. In the unsupervised framework, the proposed ConvNet model minimizes an ACWE (Chan and Vese, 2001) based energy function, which solely depends on the intensity statistics of the given image. The parameters of the ConvNet are optimized using a training set of images. The network learns a universal representation that enables the segmentation of an unseen image based on its intensity statistics. In the presence of ground truth data, we leverage the available segmentation labels during network training to incorporate structural information. We evaluate the proposed method on the task of segmentation of bone structures in 2D slices of simulated SPECT. However, the proposed model can be easily extended to 3D, and it can also be readily applied to other imaging modalities.

## 2. Methods

Fig. 1 shows an overview of the proposed method. The ConvNet takes a 2D transaxial slice of a 3D SPECT image as the input, and it outputs a mask. In an unsupervised setting, only the ACWE loss between the predicted mask and the input image is backpropagated to update the parameters. In the situation where the ground truth label exists, the labeling loss can also help to find the optimal parameters of the ConvNets.

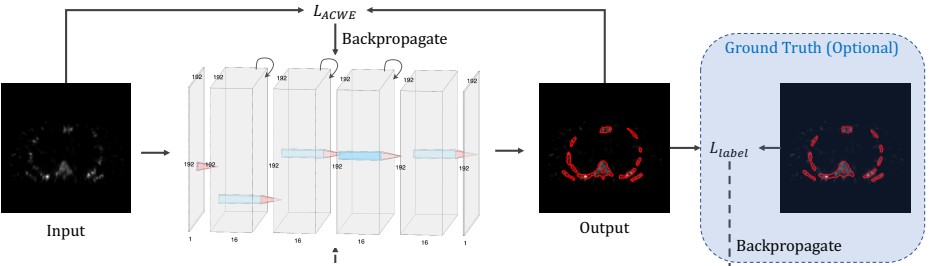

**Figure 1:** Overview of the proposed method.

**ConvNet Architecture** The network is built on the basis of the Recurrent convolutional neural network (RCNN) proposed by Liang et al. (Liang and Hu, 2015). It is a small network that consists of three recurrent and two regular convolutional layers. For each recurrent layer, $T = 3$ time-steps where used, resulting in a feed-forward subnetwork with a depth of $T + 1 = 4$ (Liang and Hu, 2015). Each convolutional layer is followed by batch normalization and a Parametric rectified linear unit (PReLU) (He et al., 2015). Notice that we continue to use PReLU in the final layer. Instead of assigning probabilities to pixel locations using Softmax, we classify any location with a value larger than 0 as foreground, and vice versa. Based on our empirical experiments, we found this was better than using Softmax, especially in the final layer, in the case of binary-class segmentation. Finally, input and output to the network share the same dimension of $192 \times 192 \times 1$.

**Loss Function** We propose to use the famous Chan and Vese model (Chan and Vese, 2001) as an unsupervised loss for ConvNet-based image segmentation. This model relies solely on the intensity statistics, which are independent of ground truth labels. It is expressed as:

$$F(c_1, c_2, C) = \mu \cdot \text{Length}(C) + \nu \cdot \text{Area}(inside(C))$$
$$+ \lambda_1 \int_{inside(C)} |g_i - c_1|^2 di + \lambda_2 \int_{outside(C)} |g_i - c_2|^2 di, \tag{1}$$

where $i$ is the index of pixel locations, $c_1$ and $c_2$ are referred to as the averages of image $g$ inside and outside the contour, $C$, and $\mu$, $\nu$, $\lambda1$, and $\lambda2$ are weighting parameters for each term. Since $\text{Length}(C)$ is in some sense comparable to $\text{Area}(inside(C))$(Chan and Vese, 2001), and for simplicity, we choose $\mu$ to be 0. Both $\lambda1$ and $\lambda2$ are chosen to be 1. The contour/segmentation is modeled by a ConvNet, $f$, with parameters $\theta$ and the input image $\mathbf{g}$, i.e., $C = f_\theta(\mathbf{g})$. The loss function can be written as:

$$\mathcal{L}_{ACWE} = \nu \cdot \text{Area}(f_\theta(\mathbf{g}) > 0) + \sum_{f_\theta(\mathbf{g}) > 0} |\mathbf{g} - c_1|^2 + \sum_{f_\theta(\mathbf{g}) \leq 0} |\mathbf{g} - c_2|^2. \tag{2}$$

In the case where segmentation labels are available for training, an optional supervised loss can be incorporated to further improve the accuracy, as shown in Fig. 1. The loss

has a similar form as $\mathcal{L}_{ACWE}$, but it is evaluated between segmentation labels (Chen et al., 2019b):

$$\mathcal{L}_{label} = \sum_{f_\theta(\mathbf{g})} |\nabla(f_\theta(\mathbf{g}))| + \sum_\Omega ((\mathbf{1} - \mathbf{u})^2 - (\mathbf{0} - \mathbf{u})^2) f_\theta(\mathbf{g}), \tag{3}$$

where $\Omega$ is the image domain, and $\mathbf{u}$ represents the ground truth labels. A weighting parameter, $\alpha$, was used to combine the two losses, i.e., $\mathcal{L} = \mathcal{L}_{ACWE} + \alpha \mathcal{L}_{label}$. We set $\alpha = 0.4$, and $\nu$ in the $\mathcal{L}_{ACWE}$ to be 0.004 based on the empirical experiments.

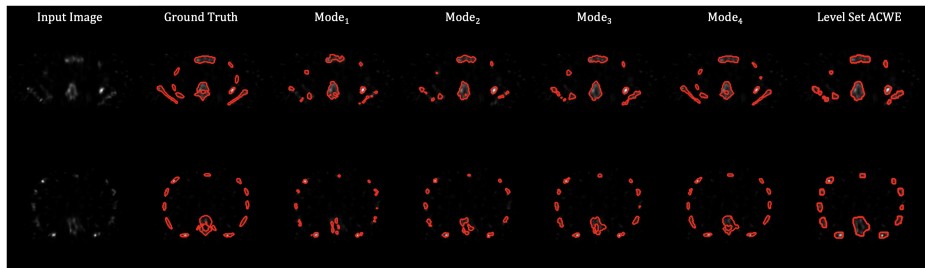

**Figure 2:** Qualitative results.

## 3. Results & Conclusions

|  | $Mode_1$ | $Mode_2$ | $Mode_3$ | $Mode_4$ | Level Set ACWE |
|---|---|---|---|---|---|
| DSC | 0.593±0.19 | 0.661±0.16 | 0.732±0.12 | 0.856±0.09 | 0.518±0.337 |

**Table 1:** DSC comparisons for different modes of the proposed method and level-set-based ACWE.

|  | Proposed Method | Level Set ACWE |
|---|---|---|
| Time (Sec) | 0.006 ± 0.022 | 2.698±0.085 |

**Table 2:** Comparisons of computational time for the proposed method and level-set-based ACWE.

We tested the proposed model on the task of bone segmentation using highly realistic simulated $^{99m}$Tc SPECT images. The object was generated based on the realistic XCAT phantom (Segars et al., 2010). We used an analytic projection algorithm that realistically models attenuation, scatter, and collimator-detector response (Frey and Tsui, 1993; Kadrmas et al., 1996). Tomographic image reconstruction was done using the ordered subsets expectation-maximization algorithm (OS-EM) with 2 and 5 iterations. A total of 140 3D volumes were simulated from 50 noise realizations. We sampled 8000 2D slices from those volumes for training the ConvNet and 4000 slices for testing and evaluation. We evaluated four settings of the proposed algorithm:

- $Mode_1$: Unsupervised (self-supervised) training with $\mathcal{L}_{ACWE}$.
- $Mode_2$: $Mode_1$ + fine-tuning using $\mathcal{L}_{label}$ with 10 ground truth (GT) labels.
- $Mode_3$: $Mode_1$ + fine-tuning using $\mathcal{L}_{label}$ with 80 GT labels.
- $Mode_4$: Training with $\mathcal{L}_{ACWE} + \mathcal{L}_{label}$.

We also compared the proposed method to the level-set-based ACWE (Chan and Vese, 2001). Although segmenting bone from SPECT images is challenging, the proposed algorithm performed well. Qualitative results, Dice coefficient (DSC) evaluations, and computational time comparisons are shown in Table 1, Table 2, and Fig. 2. As visible from the

results, fine-tuning the pre-trained unsupervised model with only 80 GT labels leads to a significant improvement in performance. In conclusion, we present an unsupervised/semi-supervised ConvNet-based model for image segmentation that can be trained with or without ground truth labels. The resulting DSC values reported demonstrate the effectiveness of the proposed method.

## Acknowledgments

This work was supported by a grant from the National Cancer Institute, U01-CA140204. The views expressed in written conference materials or publications and by speakers and moderators do not necessarily reflect the official policies of the NIH; nor does mention by trade names, commercial practices, or organizations imply endorsement by the U.S. Government.

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

## Appendix A. Qualitative Results

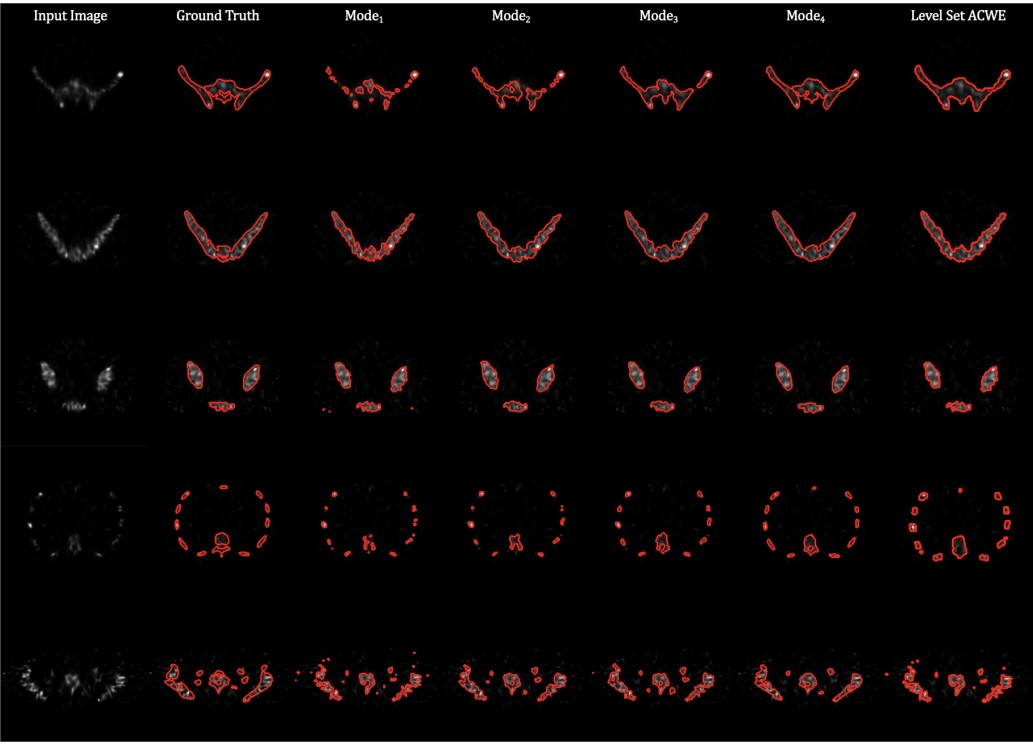

**Figure 3:** Additional results showing bone segmentation by four different settings of the proposed method.

