# OpenReview forum: "Medical Image Segmentation via Unsupervised Convolutional Neural Network"
_MIDL.io/2020/Conference — MIDL 2020_

### Official Review · AnonReviewer2 · 2020-03-09
**Merging active contours without edges and deep learning based segmentation.**

**Rating:** 4
**Confidence:** 4

**Review:**

The authors propose a recurrent CNN architecture with a new loss inspired from the  mumford shah  / chan-vese functional. Doing so, the network learns to maximize the separation between  foreground and  background in a fully unsupervised fashion. With few modifications, the approach is also adapted to the supervised case where segmentation labels are available.  The authors validate the approach on simulated phantom SPECT data.
The paper is technically sound and convincing. The idea of bringing the ACWE formalism to deep learning based segmentation is refreshing and in itself is a sufficient contribution to the field that is worth being communicated to the community.

On the negative side, the validation on simulated data is not very impressive.  Visual results seem to suggest that foreground to background separation is quite easy for this data with almost uniform black background. There are also too many typos for a 3 page paper (avaiable, prposed,..) One could also wonder how well the supervised ACWE loss compare to conventional segmentation losses.

---

### Official Review · AnonReviewer3 · 2020-03-10
**A method not suited for many medical applications**

**Rating:** 2
**Confidence:** 5

**Review:**

This paper describes a method to leverage unlabelled images to improve image segmentation via convolutional neural network.
The idea is based on the well-known “active contour without edges” segmentation method introduced by Chan & Vese, which consists in minimizing an energy such that both the background and the segmented object have homogeneous intensities, and the boundary between them is smooth.
The authors train a network with such a loss for all unlabelled images, and a standard segmentation loss for the labelled ones.

* The main limitation of this method is that it assumes that the object and respectively the background have consistent intensities, i.e. that they can be approximated by a single intensity value: $c_1$ (resp. $c_2$).
This is a very strong hypothesis that is not discussed at all by the authors.
In particular, it rarely holds in medical imaging where structures are more often distinguishable by their shape, their texture or their surrounding but not necessarily by a single and absolute intensity value.
This is actually why methods like Chan & Vese have fallen out of fashion in our field. Here this might even be more dramatic if $c_1$ and $c_2$ are supposed to represent the reference intensities for all unlabelled images.
* The experiments are based only on simulated data, and in particular on one phantom. It is also not clear whether training and validation images are really different.

On the method itself:
* I find it a bit surprising to define F with both the length and area, only to discard the length on the very next line. While I agree that the two quantities are related, they favor different kind of shapes.
Moreover, length is not that difficult to encode as a loss, for instance consider a norm of the gradient of the network output.
This is all the more surprising that this has been done in $L_{label}$, see first term of equation 3.
* Why not use the Dice coefficient or the cross entropy for the labelled images, which are widely considered to be the standard losses for medical image segmentation?

Minor issues:
* The images are not very readable, especially Figures 2 and 3.
* The statement “generating annotated data [..] could take several months to a year to complete” seems a bit exaggerated.
* The paper could benefit from a proof-reading since there are many typos, for instance
- base on -> based on
- prposed -> proposed
- CovNet -> ConvNet
- avaiable -> available
$\lambda1$ -> $\lambda_1$

---

### Official Review · AnonReviewer1 · 2020-03-10
**Interesting approch, needs some analysis**

**Rating:** 2
**Confidence:** 5

**Review:**

The authors presented an unsupervised learning approach for segmenting bones in artificial SPECT images. A recurrent neural network is used to produce a binary segmentation. The model is trained using a loss derived from the Chan and Vese active contours model. As such, it does not require manual segmentations for training. Authors also introduced an additional loss to use when ground truth labels are available, to train the model in a semi-supervised way. The experimental evaluation is performed on a series of simulates SPECT images to segment bones, in a sort of ablation study in which they trained the model in an unsupervised way, fine-tuning the model with ground truth labels and in a semi-supervised way. Results indicate that the best results are obtained using the semi-supervised approach.

Pros:
* Modelling an active contour approach using neural networks is definitely a promising line of research, specially for application in which active contours have proven to be useful (e.g. vessel segmentation in CT scans).

Cons:
* I am not sure if the proposed approach would be applicable to other problems. In the simulated SPECT images used in the paper it is clear that the background class is definitely black, and that the target class has a mean value higher than that. Then the loss function seems appropriate, because that is the most contrastive statistic between the two classes. In other problems it might be more difficult than that. It would be nice if the authors can at least ellaborate on how to extrapolate the method to other more challenging segmentation problems. Perhaps crafting new features might be a solution, as long as the computation of the features is differentiable.
* The paper lacks a comparison between the proposed approach and another simple baseline (e.g. region growing or even Otsu thresholding). Since the results of the unsupervised model are not so accurate (Mode 1 in Table 1), it is definitely necessary to analyze them in the context of other unsupervised segmentation methods.
* Using means and stds of DSC does not give us a full picture of the distribution of the DSC values. Please, replace Table 1 by a box plot.
* The paper includes some statements that are not supported by references or experiments, or that are quite hard. In my opinion this is probably due to the lack of a more in-depth
revision of the text. I would recommend the authors to double check the following sentences:
--> The statement "several months to a(n) year" is quite relative. Depending on the target problem, segmenting an image might be much easier to do.
--> "Solely rely on the statistics of intensities in a given image". Most of the segmentation methods are based only on the intensities in the image! I wouldn't pose this as a disadvantage of the method. It would be different if you mention for instance the fact that the image features have to be manually crafted.


Questions:
* What is the motivation of using a RNN instead of a classical U-Net? U-Nets are know to require not so many training images, which is relevant in the context of pushing towards an unsupervised segmentation approach.
* Using a PReLU activation as the final activation function of the network seems quite odd. Could you please elaborate a little bit more about this decision? Did you try using a sigmoid function? Is it related with the fact that the loss function requires to have a binary segmentation to compute the intensity statistics?
* Is the loss stable during training? I'd like to see the evolution of the training/validation losses per iteration or epoch.


Some other minor comments:
* Avoid repetitions in the text (e.g. "methods" and "method" in line 9 page 1). Statements like "A great deal" should be avoided as well.
* The use of English can be improved, perhaps with the help of a native English speaker.

---

### Official Review · AnonReviewer4 · 2020-03-12
**A novel active contour detection based self-supervised segmentation method with good performance on simulated data**

**Rating:** 4
**Confidence:** 4

**Review:**

Summary:

Active contour based object detection strategy is transformed into unsupervised/self-supervised learning setting for segmentation tasks. This work proposes to parameterise contour evolution with a convolution neural network and self-supervise the learning with intensity based statistics without requiring any concrete labels. A strategy to incorporate few label to further refinement segmentation is also proposed.

Strengths:
+ Use of Active contour without edges (ACWE) strategy for unsupervised/self-supervised learning is novel.
+ Further, use of intensity level statistics for self-supervision is an interesting contribution.
+ The possibility of refining segmentations with few labels is additional advantage
+ Results are convincing

Weakness:
- Perhaps due to the limitation in space, the concept of ACWE is not clearly elucidated. As the work is heavily dependent on the ideas from Chan and Vese, 2001, strengthening this discussion with further motivation is recommended
- The experiments are demonstrated on simulated data. How realistic are these images and how would the model fare on real data?
- No baseline methods are reported to appreciate the reported performance

---

### Meta-Review · Area_Chair1 · 2020-04-06
**MetaReview of Paper38 by AreaChair1**

**Rating:** 3

**Metareview:**

2 out of 4 reviewers suggested weak acceptance of this work, while the other 2 suggested weak rejection. However, most of them acknowledged that the method introduces some interesting methodological insights (by combining active contour models with deep neural networks) and the main critic seems to be related to the lack of validation in non-simulated data. Since this is a short paper, even if the experimental validation is not as strong as it could be, I'm inclined to think that this work should be accepted for publication since it introduces a novel idea for the MIDL community.

Note that the reviewers have expressed many constructive comments which could improve the quality of the final manuscript. Please, take them into account when submitting the camera ready version.

**Paper Type:**

methodological development

---

### Decision · Program_Chairs · 2020-04-11

Accept